# Meta-Analysis of Extracellular Matrix Dynamics after Myocardial Infarction Using RNA-Sequencing Transcriptomic Database

**DOI:** 10.3390/ijms232415615

**Published:** 2022-12-09

**Authors:** María Ortega, César Ríos-Navarro, Jose Gavara, Elena de Dios, Nerea Perez-Solé, Victor Marcos-Garcés, Antonio Ferrández-Izquierdo, Vicente Bodí, Amparo Ruiz-Saurí

**Affiliations:** 1INCLIVA health Research Institute, 46010 Valencia, Spain; 2Centro de Biomateriales e Ingeniería Tisular, Universidad Politécnica de Valencia, 46022 Valencia, Spain; 3Department of Medicine, University of Valencia,46010 Valencia, Spain; 4Centro de Investigación Biomédica en Red (CIBER)-CV, 28029 Madrid, Spain; 5Cardiology Department, Hospital Clínico Universitario, 46010 Valencia, Spain; 6Department of Pathology, University of Valencia, 46010 Valencia, Spain; 7Pathology Department, Hospital Clínico Universitario, 46010 Valencia, Spain

**Keywords:** myocardial infarction, extracellular matrix, RNA-sequencing, meta-analysis

## Abstract

Extracellular matrix (ECM) changes after myocardial infarction (MI) need precise regulation, and next-generation sequencing technologies provide omics data that can be used in this context. We performed a meta-analysis using RNA-sequencing transcriptomic datasets to identify genes involved in post-MI ECM turnover. Eight studies available in Gene Expression Omnibus were selected following the inclusion criteria. We compare RNA-sequencing data from 92 mice submitted to permanent coronary ligation or sham, identifying differentially expressed genes (*p*-value < 0.05 and Log2FoldChange ≥ 2). Functional enrichment analysis was performed based on Gene Ontology biological processes (BPs). BPs implicated in response to extracellular stimulus, regulation of ECM organization, and ECM disassembly were detected soon after ischemia onset. ECM disassembly occurred between days one to seven post-MI, compared with ECM assembly from day seven onwards. We identified altered mRNA expression of 19 matrix metalloproteinases and four tissue inhibitors of metalloproteinases at post-infarcted ECM remodeling and altered transcriptomic expression of 42 genes encoding 26 collagen subunits at the fibrotic stage. To our knowledge, this is the first meta-analysis using RNA-sequencing datasets to evaluate post-infarcted cardiac interstitium healing, revealing previously unknown mechanisms and molecules actively implicated in ECM remodeling post-MI, which warrant further validation.

## 1. Introduction

The extracellular matrix (ECM) is a dynamic non-cellular three-dimensional network of fibers, glycosaminoglycans, proteoglycans, and adhesion glycoproteins. The ECM provides support to tissue and helps regulate diverse cellular functions, including proliferation, survival, differentiation, migration, and inflammation [1,2,3].

Myocardial infarction (MI) is caused by the thrombotic occlusion of a coronary artery, which requires rapid reperfusion to reestablish nutrient and oxygen supply to the myocardium [4]. After MI, dynamic changes at the cellular level, as well as in ECM composition, are necessary to preserve tissue homeostasis and form a proper fibrotic scar composed mainly of collagen [5,6]. However, these alterations must be perfectly regulated in time and location to avoid adverse left ventricular remodeling and cardiovascular events. In this scenario, matrix metalloproteinases (MMPs), a group of proteases with catalytic function, and tissue inhibitors of MMP (TIMP) play a key role via the degradation of the damaged ECM components [3,5,7].

Next-generation sequencing technologies have revolutionized biomedical research by providing a wealth of omics data that can be used to elucidate the molecular mechanisms underlying diseases. Although several studies have been conducted utilizing RNA-sequencing technology on myocardium isolated from experimental models of MI [8,9,10,11,12,13], few are focused on the dynamic alterations of genes involved in ECM composition and regulation. Furthermore, these studies have been performed at different time points and with a limited number of animals.

Given the importance of a comprehensive understanding of the molecular pathways implicated in ECM remodeling and its composition after MI, as well as the vast amount of transcriptomic data generated in previously published RNA-sequencing studies, our aim in this study is to perform a quantitative meta-analysis using RNA-sequencing transcriptomic datasets focused on scrutinizing the changes in the expression of key genes implicated in ECM composition and regulation after MI induction.

## 2. Results

### 2.1. Characteristics of the Selected Studies

Figure 1 illustrates the flow chart used to select the studies in the present meta-analysis [8,9,10,11,12,13]. Briefly, 15,337 studies were found using the search term “myocardial infarction,” from which only 143 datasets were retrieved from Gene Expression Omnibus (GEO) and reviewed for eligibility. After removing duplicate series (n = 51), 84 were excluded for the following reasons: knock-out mice (n = 24), single cell or single nuclear (n = 28), evaluation of non-infarcted area (n = 11), mice submitted to pharmacological treatment (n = 14), new-born mice (n = 3), insufficient data about the dataset information (n = 2), and mice submitted to ischemia followed by four days of ischemia (n = 2).

Finally, eight datasets (including 92 animals) were employed in the meta-analysis. Animals were divided into sham mice (n = 30, without MI induction), and 62 submitted to different times of coronary ischemia: 6 h (n = 8), one day (n = 16), three days (n = 10), seven days (n = 7), 14 days (n = 10), and 21 days (n = 11). Mice undergoing more than 21 days of ischemia were included in the 21 days group. Table 1 summarizes the selected datasets, the associated publication, and the number of samples for each study group. After performing a principal component analysis, one was discarded from the meta-analysis due to differences in behavior from the other included studies.

### 2.2. GO Enrichment Analysis

When compared to sham, the following number of differentially expressed genes (DEGs) was detected in each MI group: 6447 (6 h), 14,246 (1 day), 14,354 (3 days), 13,869 (7 days), 9449 (14 days) and 12,129 (21 days). Afterward, functional enrichment analysis was carried out with the biological process (BPs) terms of gene ontology (GO) (Figure 2A).

At all ischemic times, we observed a clear upregulation (with an adjusted *p*-value lower than 0.01) of the following ECM-related BPs: (i) Extracellular structure organization (GO:0043062), (ii) ECM organization (GO: 0030198), (iii) Protein localization to the extracellular region (GO: 0071692), and (iv) Establishment of protein localization to the extracellular region (GO: 0035592).

Although BPs involved in response to the extracellular stimulus were upregulated at the initial stages of ischemia (6 h and one day after MI induction), ECM organization started in the 1 day group, as reflected by overexpression of genes implicated in the Regulation of ECM organization (GO:1903053) and Regulation of ECM disassembly (GO: 0010715) BPs. A shift in ECM behavior took place at three days of ischemia, as indicated by the appearance of new BPs, such as ECM disassembly (GO: 0022617), which remained upregulated up to day seven of ischemia. Conversely, the ECM assembly (GO: 0085029) BP showed overrepresentation in the seven days group and remained elevated in the 21 days group (Figure 2B).

To further characterize the process of post-MI matrix turnover, Figure 3 and Table 2 illustrate the name and main functions of the overexpressed genes included in the Extracellular matrix assembly (GO:0085029) and Extracellular matrix disassembly (GO:0022617) BPs. We detected that the mRNA expression genes implicated in ECM disassembly was upregulated from day one to day seven after MI induction, whereas the mRNA expression of genes involved in ECM assembly were heightened from day seven to day 21 after coronary occlusion. This tendency was also reflected by the 41 genes related to the BP of interest, which remained overexpressed from day seven until day 21 post-MI (Figure 2B). Contrarily, only six new overrepresented genes appeared on day 14, indicating that no substantial changes in the mRNA expression of genes involved in ECM remodeling appeared after seven days of ischemia (Figure 2B).

### 2.3. Dynamic Changes in Genes Encoding MMPs and TIMPs in the Infarcted Area Post-MI

Having evaluated the main BPs participating in ECM remodeling, we next sought to pinpoint the dynamic alterations in MMP and TIMP expression detected in the infarcted myocardium (Figure 4 and Figure 5, respectively). According to our data, 19 MMPs and 4 TIMPs had altered mRNA expression due to ischemic insult.

*Mmp8* and *Mmp9* became overexpressed very soon after coronary occlusion (6 h), peaking at day one of ischemia with a Log2FoldChange around 4 and returning to control levels at day seven of ischemia. The mRNA expression of *Mmp10*, *Mmp19*, *Mmp24*, and *Mmp25* was also elevated in the 6 h group, but overexpression persisted until late phases (21 days) after MI, displaying a Log2FoldChange no higher than 4.

The augmented mRNA expression of the second group of MMPs, including *Mmp16*, *Mmp17*, and *Mmp27*, started at days three to seven after ischemia onset and remained elevated at day 21. In contrast, transcriptomic levels of *Mmp11*, *Mmp23*, and *Mmp28* exhibited a bimodal pattern: their mRNA expression was downregulated in the first 24 h after ischemia, whereas significant upregulation was observed in the infarcted tissue isolated from day seven to day 21 post-MI. Lastly, *Mmp15* gene expression followed a different pattern, showing a significant decrease in gene expression after the ischemic insult.

Regarding the endogenous inhibitors of MMPs, an altered mRNA expression of 4 different TIMPs were observed. Concretely, a clear upregulation of *Timp1* was detected at all ischemic times, while *Timp2* and *Timp3* mRNA expression only increased from day seven onwards. On the contrary, the *Timp4* gene expression was downregulated after coronary occlusion (Figure 5).

Overall, 19 different MMPs and 4 TIMPs were implicated in post-infarcted ECM healing. Of these, *Mmp8*, *Mmp9*, *Mmp10*, *Mmp19*, *Mmp24*, and *Mmp25* displayed early overrepresentation during ECM disassembly, whereas a second group (*Mmp27*, *Mmp11*, *Mmp16*, *Mmp17*, *Mmp23*, and *Mmp28* and also *Timp2* and *Timp3*) were involved at late stages, when most ECM assembly takes place.

### 2.4. Dynamic Alterations in Transcriptomic Analysis of Genes Encoding Collagen Subunits

Lastly, we studied the mRNA expression of genes encoding different collagen subunits ultimately implicated in post-MI solid fibrotic scar formation (Figure 6). Based on our results, a total of 42 genes related to 26 different collagen subunits displayed altered mRNA expression within the infarct region compared to control tissue. In genes associated with collagen subunits, behavior overall was bimodal: decreased mRNA expression until the day one post-MI followed by massively increased expression (with Log2FoldChange greater than 2) from day three to chronic phases (21 days).

In total, 18 genes involved in the formation of 12 different collagen molecules displayed downregulated mRNA expression at the acute phase (until day 3) of post-infarcted ECM healing. Afterward, they showed significantly enhanced expression at subacute phases (7–14 days after coronary occlusion) compared to control myocardium. Contrarily, *Col6a6*, *Col13a1*, and *Col25a1* genes showed no augmentation in mRNA expression at chronic phases of ischemia.

The highest transcriptional expression of genes involved in collagen formation was detected on day seven after coronary ligation. Of note, *Col1a1*, *Col1a2*, *Col2a1*, *Col3a1*, *Col8a1*, *Col8a2*, *Col9a1*, *Col9a2*, *Col11a1*, *Col12a1*, *Col14a1*, and *Col24a1* displayed a Log2FoldChange of 5–9 on day seven of ischemia, a tendency that persisted up to day 21. Lastly, in line with previous data, collagen subunits showed similar expression at day 14 to those overrepresented at day 21 after ischemia onset, where 34 genes involved in the formation of 22 collagen subunits displayed heightened mRNA expression.

Altogether, alterations in the transcriptomic expression of up to 26 different collagen subunits were detected along the process of post-infarcted ECM remodeling. The maximum number of upregulated genes occurred in the seven days group, in parallel with onset of ECM assembly. This tendency was sustained at the chronic phase (21 days after coronary occlusion).

## 3. Discussion

This study aims to provide insight into variations in the cardiac interstitium occurring after MI via designing a quantitative meta-analysis of eight independent RNA-sequencing studies using infarcted tissue isolated at various time points after ischemia onset. Specifically, our results conclude that the augmented mRNA expression of genes implicated in ECM disassembly occurs within the first seven days after MI, while upregulation of ECM assembly occurs from day seven onwards. Using this novel approach, we have identified up to 19 different MMPs and 4 TIMPs participating in post-infarcted ECM turnover and 26 different collagen subunits displaying altered transcriptomic expression during fibrotic scar formation (Figure 7). This new data could be of interest for designing future clinical and experimental approaches to shed light on the pathophysiology of MI, specifically on ECM remodeling post-MI.

### 3.1. Role of Cardiac Interstitium Following MI

MI, one of the main causes of morbimortality in western countries, is caused by the thrombotic obstruction of a coronary artery, leading to a lack of oxygen and nutrients in the downstream myocardium [4,14]. Ischemic insult might provoke massive cardiomyocyte necrosis and ECM alterations which must be sequentially resolved in time and location. Firstly, an orchestrated immune reaction is activated to clean the infarcted myocardium from damaged cells and ECM components. Following this, myofibroblast-secreted collagen fibers are decisive to constitute a solid scar [5,15,16]. Cardiac ECM, a three-dimensional network of organized macromolecules responsible for providing biomechanical and biochemical support to cardiomyocytes, also experiences substantial alterations after MI [5,17]. In fact, as exaggerated cardiac ECM remodeling can cause the development of post-MI heart failure, optimal fibrotic scar formation is of utmost importance. Therefore, deeper comprehension of post-infarcted ECM changes and key molecules implicated in this process might provide the key to flawless management of cardiac reparation.

### 3.2. RNA-Sequencing Technology to Further Comprehend Mechanisms Underlying ECM Changes

Omics technologies have revolutionized biomedical research in recent years, with the emergence of a wealth of data regarding different physiological and pathological situations. Although several studies have been conducted utilizing RNA-sequencing technology on infarcted myocardium isolated from experimental models [8,9,10,11,12,13], there is a lack of publications focused on ECM, a central and sometimes inconspicuous actor in the pathophysiology of MI. For this study, we therefore, designed a meta-analysis of previously published RNA-sequencing studies, focusing on elucidating the alterations in the expression of the principal genes implicated in ECM composition and regulation after MI.

Our meta-analysis included data from eight studies in which infarcted mice hearts were evaluated via RNA-sequencing analysis. The novelty of our study lies in the following: (i) no previous publications have examined the mechanisms underpinning ECM healing after MI; (ii) our substantial sample size (n = 92), including tissue isolated at different time points after ischemia onset, allows us to characterize their dynamics, and also strengthens the statistical power of our results.

Based on the results of our analysis of gene expression and functional enrichment on DEGs, communication between damaged cardiac interstitium and heart resident cells has already begun 6 h after coronary occlusion, resulting in overexpression of BPs related to Regulation of ECM disassembly and Regulation of ECM organization 24 h post-MI. Upregulation of genes implicated in ECM disassembly was detected in the three days group. A previous study from Yan et al. demonstrated that neutrophils and M1-like macrophages are predominantly present in the infarcted myocardium at this specific time point [18]. Both leukocyte subtypes exert pro-inflammatory functions and are reported to actively participate in the clearance of injured ECM. At day seven post-MI, overexpression of the BPs implicated in not only ECM disassembly but also ECM assembly takes place, illustrating the intriguing and dynamic behavior of the cardiac interstitium. Finally, ECM protein deposition persists until chronic phases.

Collectively, these results indicate that rapid genetic changes occur soon after coronary occlusion in an orchestrated fashion aimed at activating molecular pathways to clean and repair infarcted cardiac interstitium. Moreover, ECM disassembly occurs within the first seven days after MI induction, whereas ECM assembly occurs from day seven onwards.

### 3.3. Alterations in MMP and TIMP Expression after MI

In post-infarction healing, MMPs (a group of 25 proteolytic enzymes) are expressed by various cell types and are mainly responsible for ECM turnover, cytokine signaling, and leukocyte cell trafficking. MMPs activity is controlled by specific inhibitors, named TIMPs, which are also involved in angiogenesis, cell proliferation, and apoptosis [19,20]. Scrutinizing the existence of both MMPs per se and peptide fragments generated by MMP cleavage is crucial to advance the understanding of post-MI heart failure development and hasten the discovery of potential predictive biomarkers [21,22]. In the current study, we focused on evaluating the dynamic variations in mRNA expression of genes encoding MMPs in the infarcted myocardium after ischemic insult.

According to our data, the mRNA expression of 19 MMPs was altered post-MI, showing a bimodal pattern. MMP8, MMP9, MMP10, MMP19, MMP24, and MMP25 displayed an early overrepresentation during ECM disassembly, whereas MMP27, MMP11, MMP16, MMP17, MMP23, and MMP28 are involved at late stages (during ECM assembly). Although further experiments are needed, a plausible explanation might be that the first group of MMPs participates in removing damaged ECM components, while the second is involved in developing a solid fibrotic scar. In fact, this hypothesis is reinforced by the results obtained regarding the transcriptomic expression of TIMPs post-MI. The clear augmentation of three different TIMPs (TIMP1, TIMP2, and TIMP3) from day seven onwards might participate in the inhibition of MMPs, thus reducing ECM disassembly and starting ECM assembly. These data are in line with those previously reported in a swine MI model since mRNA levels of TIMP1, TIMP2, and TIMP3, but not TIMP4, were increased in the necrotic area of hearts submitted to 90 min of ischemia followed by seven days of reperfusion [3].

Regarding the implication and dynamics of MMPs in ECM healing, MMP2, MMP3, MMP8, MMP9, MMP13, MMP14, and MMP28 have been widely described in the current literature, and results are in line with those obtained in this meta-analysis [21,23,24,25,26,27]. In contrast, the role of MMP10, MMP11, MMP15, MMP16, MMP17, MMP18, MMP23, MMP24, MMP25, and MMP27 post-MI has more recently been investigated via immunoblotting and immunohistochemistry [21], showing slight discrepancies with our results. This disagreement is most likely due to variations in study design: Kaminski et al. determined mRNA expression only on macrophages and fibroblasts [21], while our results are based on the transcriptome of the whole cardiac tissue.

Overall, our study breaks new ground by expanding current knowledge on MMPs in this scenario: we have demonstrated the participation and dynamics of up to 19 different MMPs and 4 TIMPs in the turnover and reparation of the infarcted ECM after MI. Additionally, these results could lay the groundwork for designing future experimental approaches to further elucidate the function of new and understudied MMPs in MI scenarios.

### 3.4. Dynamic Changes in the Expression of Genes Implicated in Collagen Synthesis

Fibrotic scar formation constitutes the end result of the myocardial reparative process after different cardiac insults, including MI. Contrary to the insignificant regenerative capacity of adult cardiomyocytes, replacement fibrosis mediated by myofibroblasts clearly reflects a vigorous biological process aimed at promoting solid scar formation and preventing deleterious left ventricular remodeling [1,6,7,27]. The participation of collagen in fibrotic scar formation following MI has been widely described [1]. Traditionally, research has focused on evaluating type I and type III collagen, considered the main components of the infarcted heart at the fibrotic phase [1,3,6]. Indeed, their amount and spatial orientation is considered crucial to ensure proper mechanical and electrical functions in the repaired myocardium [28,29]. However, little is known about the pathophysiological implications of other subunits of this molecule.

Frangogiannis and colleagues recently published an in-depth review describing fibrillar and nonfibrillar collagens present in cardiac tissue after MI or ischemic heart failure [1]. According to their data, no information was available on the role of 14 collagen subunits in this scenario (i.e., Col VII, VIII, IX, X, XII, XIII, XVI, XVII, XXII, XXIII, XXIV, and XXV) [1]. Nonetheless, developing transcriptomic and proteomic techniques have brought promising results to further comprehend the implications of novel collagen subunits in fibrotic scar formation.

A first study employing next-generation sequencing was performed by Yokota et al. [8] and elucidated the previously unknown role of Col VIII, XII, and XVI post-MI. Briefly, in mice undergoing permanent coronary ligation, upregulated mRNA expression of these three subunits was reported to heighten by day three, peak at day seven, and diminish by six weeks in the infarct area. These data concur with those obtained in our meta-analysis performed in almost 100 MI animals. Furthermore, our study reveals a total of 26 collagen subunits represented after coronary occlusion. Specifically, collagen-related gene transcription is demonstrated to peak at only seven days after coronary occlusion, paralleling the onset of ECM assembly. In fact, even at late-stage phases (21 days post-MI), increased mRNA expression persisted in 34 genes from 22 different collagen subunits. To our best knowledge, this is the first study illustrating the involvement of such a large quantity of different collagen subunits during post-MI cardiac fibrosis.

Nonetheless, it should be kept in mind that augmented gene expression is not always translated to the protein level. For instance, although heightened mRNA expression of Col XI and Col XXIV was noted in our meta-analysis, no protein modifications in these two subunits were demonstrated in swine and human cardiac interstitium isolated after MI [30]. Although the mRNA upregulation is not always transferred at the protein level, our study could help establish the basis for designing further clinical and experimental studies addressed at broadening the range of potential collagen subunits participating in post-MI replacement fibrosis.

### 3.5. Limitations of the Study

In our study, we could not detect DEGs among different datasets due to the statistical analysis performed.

Although the cellular component is also part of the ECM, no data is available to properly describe the changes in the cellular components after MI.

## 4. Materials and Methods

To select the studies for the meta-analysis, a dataset search using the term “myocardial infarction” was carried out in the GEO database [31]. The study search ended in September 2021, and the inclusion criteria were studies performed in *Mus musculus*, animals submitted to MI or sham, and studies with bulk RNA-sequencing analysis from the infarcted myocardium. The exclusion criteria were studies performed by single cell or single nucleus RNA sequencing, mice submitted to any pharmacological or mechanical intervention, studies with newborn mice, analyses carried out in blood or non-infarcted myocardium, and studies not correctly filtered by the GEO filters.

The datasets were evaluated following inclusion and exclusion criteria, as shown in the complete study flow chart detailed in the PRISMA Flow Diagram (Figure 1). Afterward, the SRA Run Selector was used to obtain the accessions and download the sequence data files using SRA Toolkit. The FASTA files were aligned to the reference genome (GRCm38) with Bowtie2 [32]. Read counts for each gene were calculated using hTSeq [33] to obtain expression levels.

All the following computational analyses were performed in the R environment [34]. Read counts for each study were normalized (except for the GSE83350 dataset, which only has one animal per study group), saved, and combined with the normalized counts of all studies in a unique DESeqDataSet. The design was also adjusted for batch effect (the study from which the data have been obtained). Differential expression analysis was performed with the DESeq2 (1.30.1) [35] to compare all the infarcted groups with sham. A *p*-adjusted value < 0.05 and a Log2FoldChange ≥ 2 cutoff were used for determining DEGs.

Functional enrichment analysis of the DEGs was performed with the Cluster Profiler package [36] using GO annotations [37] with the BP category and adjusting the *p*-value with the Benjamin-Hochberg method. BPs with a *p*-value less than 0.05 were selected. As we aimed to evaluate BPs related to the ECM, the descriptions were filtered using the terms “extracellular” and “matrix.” Graphical representations were carried out with the ggplot2 (3.3.5) [38], UpSetR (1.4.0) [39], and pheatmap (1.0.1) [40] packages.

## 5. Conclusions

To our knowledge, this is the first meta-analysis to use RNA-sequencing datasets to gain insight into post-infarcted ECM healing. As derived from this novel approach, genetic upregulation of the molecular mechanisms directed at repairing injured ECM start as early as 6 h after infarction induction. ECM disassembly upregulation takes place within the first seven days post-coronary occlusion, whereas genetic mechanisms addressed at promoting ECM assembly occur from day seven onwards. The mRNA levels of genes encoding 19 different MMPs, four TIMPs, and 26 collagen subunits are substantially altered within the infarct region. These novel data could lay the groundwork for designing future experimental and clinical approaches to further comprehend ECM healing in the setting of acute MI.

## Figures and Tables

**Figure 1 ijms-23-15615-f001:**
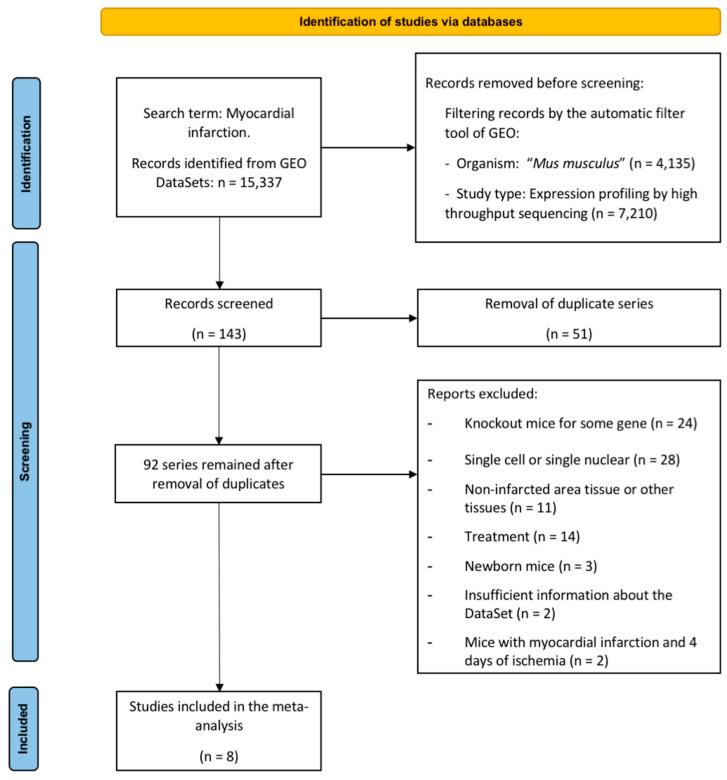
PRISMA meta-analysis flow diagram. Abbreviation. GEO: Gene expression omnibus.

**Figure 2 ijms-23-15615-f002:**
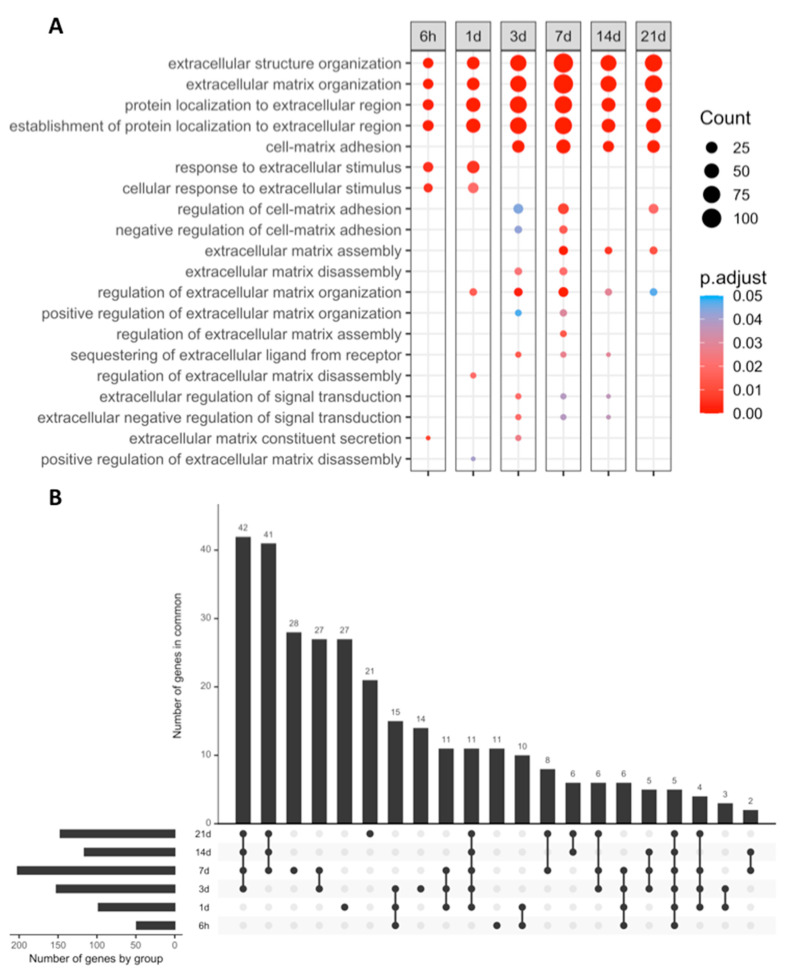
GO BPs enrichment analysis filtered by the terms “matrix” or “extracellular.” (**A**) Extracellular matrix-related BPs overrepresented at each time point after ischemia onset. Myocardial infarction groups are in the columns, and rows represent the different BPs. The size of the circle indicates the number of genes overexpressed, whereas the colors ranging from red to blue represents the *p*-value adjusted as shown in the legend. Red circles in the figure indicate lower *p*-values, while blue represent higher *p*-values. (**B**) UpSet plot of DEGs. The bottom panel shows the total number of DEGs (*x*-axis) at each time point following coronary occlusion (*y*-axis), while the upper panel shows the intersection of gene sets at multiple time points. Each column corresponds to a time point or set of time points (dots connected by lines below the *x*-axis) containing the same DEGs. The number of genes in each set appears above the column, while the time points shared are indicated in the graphic below the column, with the time points on the left. Abbreviations. BP: Biological process. DEG: Differentially expressed genes GO: Gene ontology.

**Figure 3 ijms-23-15615-f003:**
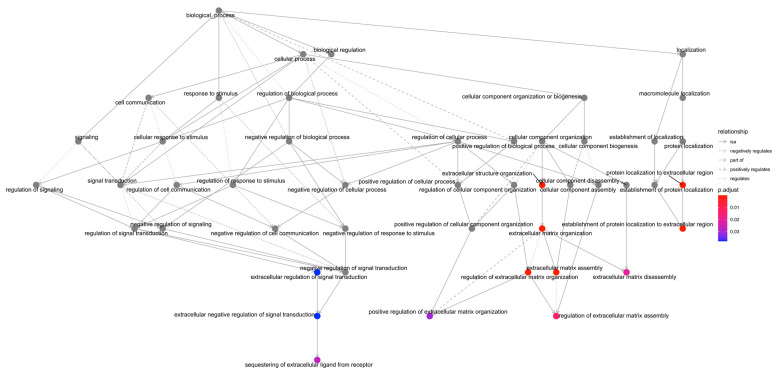
BPs enrichment map related to the terms “matrix” or “extracellular” in samples isolated from the seven days group. Hierarchical organization of GO gene-set enrichment results for the seven days group compared to control myocardium. Colors ranging from red to blue represents the *p*-value adjusted as shown in the legend. The red circles in the figure indicate lower *p*-values, while blue represents higher *p*-values. Abbreviations. BP: Biological process. GO: Gene ontology.

**Figure 4 ijms-23-15615-f004:**
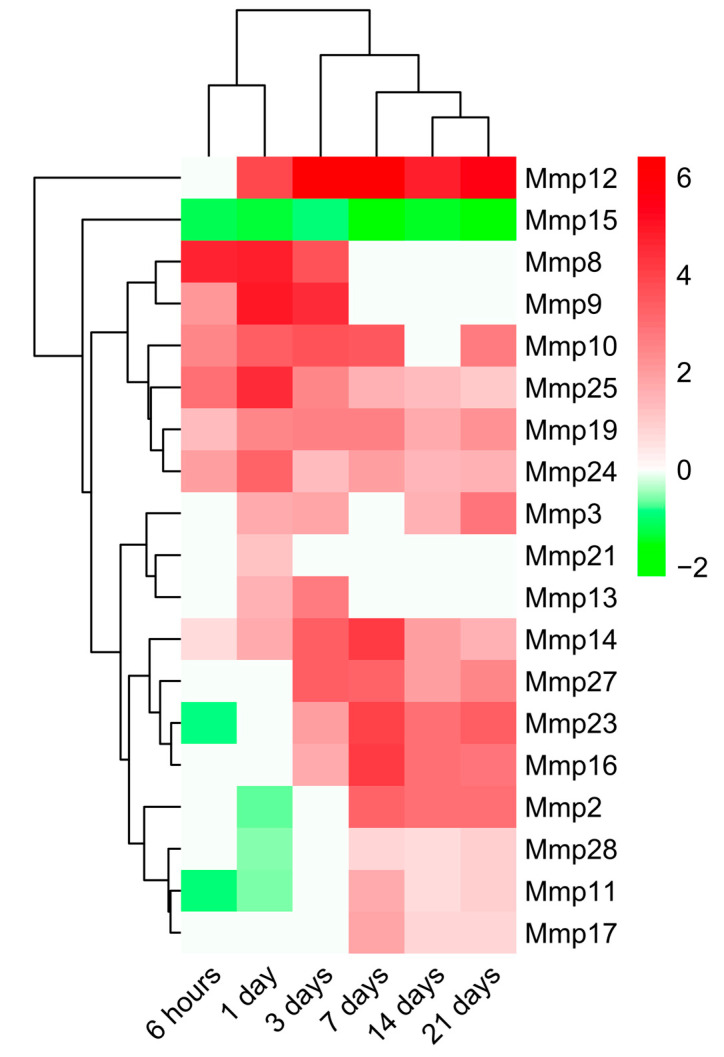
Heatmap of the Log2FoldChange of MMPs genes at different time points after coronary occlusion. Myocardial infarction groups are in the columns, and the genes encoding MMPs are in the rows. Gene intensities were log2 transformed and are displayed as colors ranging from red to green, as shown in the key. Red squares in the figure represent high-expression genes, while green squares represent low-expression genes. Both rows and columns are clustered using correlation distance and average linkage. Abbreviation. MMP: Matrix metalloproteinase.

**Figure 5 ijms-23-15615-f005:**
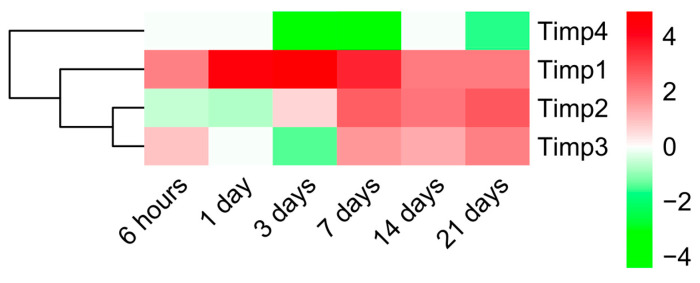
Heatmap of the Log2FoldChange of TIMPs genes at different time points after coronary occlusion. Myocardial infarction groups are in the columns, and the different genes encoding TIMPs are in the rows. Gene intensities were log2 transformed and are displayed as colors ranging from red to green, as shown in the key. Red squares in the figure represent high-expression genes, while green squares represent low-expression genes. Both rows and columns are clustered using correlation distance and average linkage. Abbreviation. TIMP: Tissue inhibitor of metalloproteinases.

**Figure 6 ijms-23-15615-f006:**
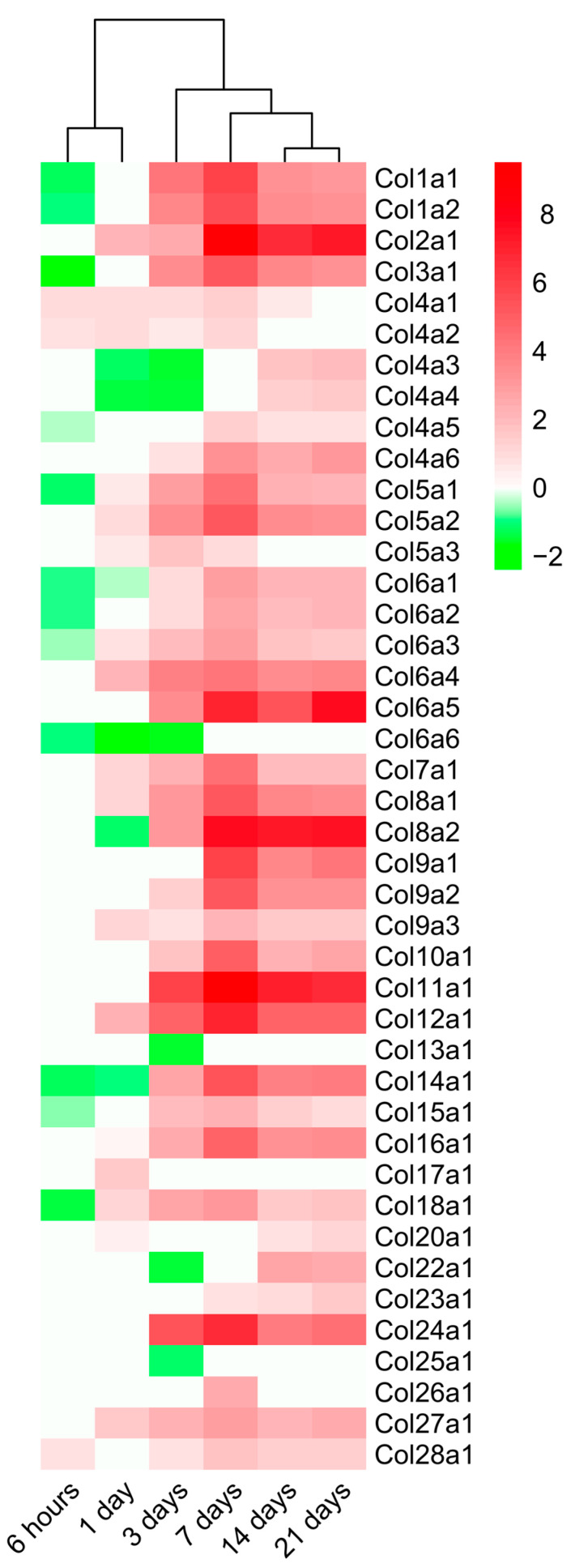
Heatmap of the Log2FoldChange of the collagen genes at different time points after coronary occlusion. Myocardial infarction groups are in the columns, and the genes encoding collagen subunits are in the rows. Gene intensities were log2 transformed and are displayed as colors ranging from red to green, as shown in the key. Red squares in the figure represent high-expression genes, while green squares represent low-expression genes. Both rows and columns are clustered using correlation distance and average linkage.

**Figure 7 ijms-23-15615-f007:**
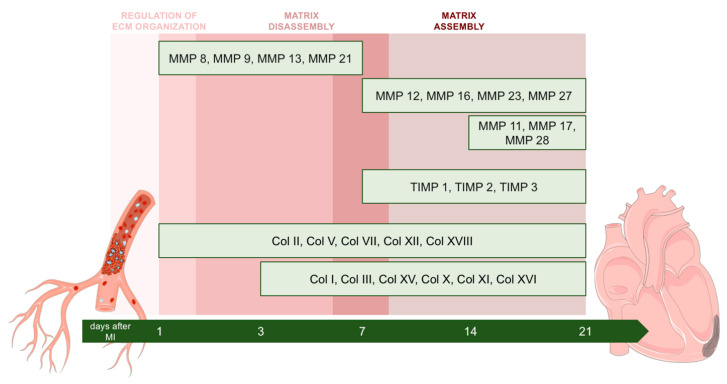
Central illustration. ECM disassembly occurred between days one to seven post-MI, compared with ECM assembly from day seven onwards. The altered mRNA expression of 19 matrix metalloproteinases and four TIMPs at post-infarcted ECM remodeling as well as altered transcriptomic expression of 42 genes encoding 26 collagen subunits at the fibrotic stage, were detected. Abbreviations. ECM: Extracellular matrix. MI: myocardial infarction. MMP: Matrix metalloproteinase. TIMP: Tissue inhibitor of metalloproteinases. Figure was produced using Servier Medical Art.

**Table 1 ijms-23-15615-t001:** Summary of GEO datasets for the meta-analysis.

GEODatasets	Reference	Number of Samples						
		Sham	6 h	1 Day	3 Days	7 Days	14 Days	21 Days
GSE153494	[9]	3	3	3	3			
GSE153493	N/A	3		3			3	
GSE151834	[8]				4	4	4	8
GSE114695	[10]	9		3		3		3
GSE153485	[11]	10	5	5				
GSE154072	N/A						3	
GSE104187	[12]	4		2	2			
GSE83350	[13]	1			1			
Total		30	8	16	10	7	10	11

Abbreviations. GEO: Gene expression omnibus. N/A: not available.

**Table 2 ijms-23-15615-t002:** Full name and function of the genes differentially expressed at different time points of ischemia and implicated in matrix disassembly and assembly (Gene Ontology biological processes).

	Gene	Name	6 h	1 Day	3 Days	7 Days	14 Days	21 Days
**MATRIX DISASSEMBLY**	** *Fgfr4* **	Tyrosine Kinase Related To Fibroblast Growth Factor Receptor		X				
	** *Il6* **	Interleukin-6		X	X	X		
	** *Pdpn* **	Glycoprotein 36		X	X	X		
	** *Tgfb2* **	Transforming Growth Factor Beta 2		X		X		
	** *Lcp1* **	Lymphocyte Cytosolic Protein 1			X	X		
	** *Fscn1* **	Fascin Actin-Bundling Protein 1			X			
	** *Ctss* **	Cathepsin S			X	X		
	** *Sh3pxd2b* **	Tyrosine Kinase Substrate With Four SH3 Domains			X	X		
	** *Mmp13* **	Matrix Metallopeptidase 13			X			
	** *Fap* **	Fibroblast Activation Protein Alpha				X		
	** *Kif9* **	Kinesin Family Member 9				X		
**MATRIX ASSEMBLY**	** *Sox9* **	SRY-Box Transcription Factor 9				X	X	X
	** *Fkbp10* **	FKBP Prolyl Isomerase 10				X		
	** *Fbln5* **	Fibulin 5				X	X	X
	** *Has2* **	Hyaluronan Synthase 2				X		
	** *Lox* **	Lysyl Oxidase				X	X	X
	** *Efemp2* **	EGF Containing Fibulin Extracellular Matrix Protein 2				X		
	** *Ltbp3* **	Latent Transforming Growth Factor Beta Binding Protein 3				X	X	X
	** *Emilin1* **	Elastin Microfibril Interfacer 1				X		
	** *Col1a2* **	Collagen Type I Alpha 2 Chain				X	X	X
	** *Phldb2* **	Pleckstrin homology Like Domain Family B Member 2				X		
	** *Antxr1* **	ANTXR Cell Adhesion Molecule 1				X	X	X
	** *Mfap4* **	Microfibril Associated Protein 4				X	X	X
	** *Gpm6b* **	Glycoprotein M6B						X

## Data Availability

The data presented in this study are available on request from the corresponding author.

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
