# Peer review of "Meta-Analysis of Extracellular Matrix Dynamics after Myocardial Infarction Using RNA-Sequencing Transcriptomic Database"

_ijms, 2022, doi:10.3390/ijms232415615_

Round 1

Reviewer 1 Report

The goal of proposed study is significant.  The results of the meta-analysis had significant value, although its limited. The paper can be improved with the following suggestions.

(1) How consistent of the differential expressed genes among different datasets? 

(2) the EMC is regulate b not only MMPs but also TIMPs.  How are the TIMPs during post-I/R remodeling?

(3) What is the transient ischemia? How long? Are they the same among different experiment?

(4) It will be more meaningful if the analysis include markers of different cell types, such as my-fibroblast and endothelial cells at different post-I/R time points and whether they correlate with the MMPs and collagens changes.

Author Response

REVIEWER 1

The goal of proposed study is significant.  The results of the meta-analysis had significant value, although its limited. The paper can be improved with the following suggestions.

(1) How consistent of the differential expressed genes among different datasets? 

This is a very interesting question. There are two types of meta-analysis using RNA-sequencing transcriptomic databases. One approach is to evaluate the differentially expressed genes between datasets in order to detect variations between the included studies. Another alternative is to unify the data derived from different datasets in a unique dataset and perform the statistical analysis taking the transcriptomic values as a single study. In order to fulfill the objectives of our study, we have performed this second approach.

The advantages of our study design are the following: i) sample size is higher, thus the statistical power of the analysis is augmented; ii) temporal changes in the transcriptomic expression of genes implicated in ECM remodeling could be assessed at various times after ischemia onset.

We agree that based on our study design, we are unable to detect how consistent the differential expressed genes among different datasets are. This is a limitation of our work. However, our objective was to perform a meta-analysis using RNA-sequencing transcriptomic datasets to identify genes involved in post-MI ECM turnover, not to detect differences among datasets.

As previously mentioned, a complete different statistical analysis are needed to achieve the suggestion of the reviewer. For that reason, we are far from answering his/her request and consequently, this issue has been mentioned in “Limitations of the study” Section:

“In our study, we are unable to detect DEGs among different databasets due to the statistical analysis performed”.

(2) The EMC is regulate by not only MMPs but also TIMPs.  How are the TIMPs during post-I/R remodeling?

We thank the reviewer for this suggestion because TIMPs are actively implicated in post-MI cardiac remodeling by inhibiting MMP function. For that reason, we have performed a re-analysis and included data regarding the alterations in TIMPs gene expression after MI. Briefly, changes in the transcriptomic expression of 4 different TIMPs were detected. Timp1 was overexpressed at all ischemic times, while Timp2 and Timp3 increased from day 7 onwards. Contrarily, the mRNA expression of Timp4 was reduced after coronary occlusion.

This new data has been specified in the following sections:

Results Section, page 8, paragraph 3:

“Regarding the endogenous inhibitors of MMPs, an altered mRNA expression of a total of 4 different TIMPs were observed. Concretely, a clear upregulation of Timp1 was detected at all ischemic times, while Timp2 and Timp3 mRNA expression were only increased from day 7 onwards. On the contrary, Timp4 gene expression was downregulated after coronary occlusion (Figure 5)”.

Discussion Section, page 12, paragraph 2:

“In fact, this hypothesis is reinforced by the results obtained regarding the transcriptomic expression of TIMPs post-MI. The clear augmentation of 3 different TIMPs (TIMP1, TIMP2 and TIMP3) from day 7 onwards might participate in the inhibition of MMPs, thus reducing ECM disassembly and starting ECM assembly. These data are in line with those previously reported in a swine MI model since mRNA levels of TIMP1, TIMP2 and TIMP3, but not TIMP4, were increased in the necrotic area of hearts submitted to 90 minutes of ischemia followed by 7 days of reperfusion [3]”.

Moreover, Figure 5 has been added.

Abstract and Conclusions were also modified to highlight the novel results obtained regarding TIMPs implication post-MI. 

(3) What is the transient ischemia? How long? Are they the same among different experiment?

In this meta-analysis, we have included transcriptomic data from different datasets. After carefully checking the manuscripts derived from GEO datasets, we realized that MI was induced by permanent coronary occlusion instead of transient coronary ligation. We apologize for this mistake because we initially thought experiments were performed in reperfused MI models. However, as previously mentioned, RNA-sequencing data are obtained in models of non-reperfused MI (permanent coronary occlusion).

For that reason, we have revised and changed the entire manuscript taking into account that MI groups are classified depending on the time of ischemia, and not time of reperfusion. Concretely, animals were divided into sham mice (n=30, without MI induction) and 62 submitted to different times of coronary ischemia: 6 hours (n=8), 1 day (n=16), 3 days (n=10), 7 days (n=7), 14 days (n=10), and 21 days (n=11). Moreover, in Results and Discussion Section, we have indicated that the temporal changes in mRNA expression are based on the time under ischemia and not after reperfusion.

Collectedly, we really appreciate this comment because it makes us realize of our mistake and apologize for that. In the revised version of the manuscript, we have made modifications through the entire manuscript to specify this issue as well as explain and discuss the results according to this information.

(4) It will be more meaningful if the analysis include markers of different cell types, such as myofibroblast and endothelial cells at different post-I/R time points and whether they correlate with the MMPs and collagens changes.

We fully agree with the reviewer comment. However, in this meta-analysis, our aim was to concretely study changes in fibers and ground substance but not the cellular composition of the ECM. The amount of data derived from this meta-analysis was huge and, for that reason, we have decided to focus on alterations in collagen fibers, MMPs and TIMPs.

In order to report changes at cellular level, single-cell RNA-sequencing is mandatory and this concrete analysis was not performed in the selected studies. We acknowledge this could be a limitation of our study since the cellular component is also an essential part of the cardiac ECM. However, there is no available data to properly obtain these results, and consequently, this issue was raised in the “Limitations of the study” Section:

“Although the cellular component is also part of the ECM, no data is available to properly describe the changes in cellular component after MI”.

Reviewer 2 Report

This study is a quantitative meta-analysis based on RNA sequencing datasets from eight independent studies, the main purpose of which is the assessment of the expression of key genes of ECM composition and remodeling following MI reperfusion and scar healing. 

The analysis sheds light on the dynamic changes of expression of several genes, identifying biological processes of degradation and reconstitution of the ECM and healing of the scar along the post-infarction remodeling of the myocardium. Notably, the study pinpoints significant temporal variations of the mRNA level of 19 different MMPs and 26 collagen subunits genes, not previously described in such details.

The ECM plays a pivotal role in the adaptive modifications and repair of the myocardium; therefore, understanding the mechanisms and the key players of the remodeling process and of the molecular interactions regulating matrix changes is of great importance in cardiovascular disease.

The precise description of the differential expression of an exhausting list of MMPs and Collagen subunits, thanks to data coming from tissues isolated at different time points from the acute to the chronic phase, after coronary reperfusion, add to the field and encourage further investigations.

Considering the data available from the present meta analysis, there are several questions that in principle can stem for discussion: for example, i) is there any interesting hints about differential expression of TIMPs, which coordinated expression with MMPs is critical for proper ECM dynamic? ii) Similarly, could it be interesting to evaluate the expression of other categories of important genes, such as growth factors or other proteins involved in matrix remodeling?

Author Response

REVIEWER 2

This study is a quantitative meta-analysis based on RNA sequencing datasets from eight independent studies, the main purpose of which is the assessment of the expression of key genes of ECM composition and remodeling following MI reperfusion and scar healing.

The analysis sheds light on the dynamic changes of expression of several genes, identifying biological processes of degradation and reconstitution of the ECM and healing of the scar along the post-infarction remodeling of the myocardium. Notably, the study pinpoints significant temporal variations of the mRNA level of 19 different MMPs and 26 collagen subunits genes, not previously described in such details.

The ECM plays a pivotal role in the adaptive modifications and repair of the myocardium; therefore, understanding the mechanisms and the key players of the remodeling process and of the molecular interactions regulating matrix changes is of great importance in cardiovascular disease.

The precise description of the differential expression of an exhausting list of MMPs and Collagen subunits, thanks to data coming from tissues isolated at different time points from the acute to the chronic phase, after coronary reperfusion, add to the field and encourage further investigations.

Considering the data available from the present meta analysis, there are several questions that in principle can stem for discussion: for example,

  1. i) is there any interesting hints about differential expression of TIMPs, which coordinated expression with MMPs is critical for proper ECM dynamic?

This is a very interesting suggestion because TIMPs are actively implicated in post-MI cardiac remodeling by inhibiting MMP function. For that reason, we have made a re-analysis and included data regarding the alterations in TIMPs gene expression after MI. Briefly, changes in the transcriptomic expression of 4 different TIMPs were detected. Timp1 was overexpressed at all ischemic times, while Timp2 and Timp3 increased from day 7 onwards. Contrarily, the mRNA expression of Timp4 was reduced after coronary occlusion.

This new data has been specified in the following sections:

Results Section, page 8, paragraph 3:

“Regarding the endogenous inhibitors of MMPs, an altered mRNA expression of a total of 4 different TIMPs were observed. Concretely, a clear upregulation of Timp1 was detected at all ischemic times, while Timp2 and Timp3 mRNA expression were only increased from day 7 onwards. On the contrary, Timp4 gene expression was downregulated after coronary occlusion (Figure 5)”.

Discussion Section, page 12, paragraph 2:

“In fact, this hypothesis is reinforced by the results obtained regarding the transcriptomic expression of TIMPs post-MI. The clear augmentation of 3 different TIMPs (TIMP1, TIMP2 and TIMP3) from day 7 onwards might participate in the inhibition of MMPs, thus reducing ECM disassembly and starting ECM assembly. These data were in line with those previously reported in a swine MI model since mRNA levels of TIMP1, TIMP2 and TIMP3, but not TIMP4, were increased in the necrotic area of hearts submitted to 90 minutes of ischemia followed by 7 days of reperfusion [3]”.

Moreover, Figure 5 has been added.

Abstract and conclusions were also modified to highlight the novel results obtained regarding TIMPs implication post-MI.

  1. ii) Similarly, could it be interesting to evaluate the expression of other categories of important genes, such as growth factors or other proteins involved in matrix remodeling?

We totally agree with this comment. For that reason, we have included in Table 2 important genes implicated in matrix remodeling, which are differentially expressed at different time points after myocardial infarction. In fact, genes were classified depending on their implication in matrix disassembly or matrix assembly. We detected that the mRNA expression genes implicated in extracellular matrix disassembly were upregulated from day 1 to day 7 after myocardial infarction induction, whereas the mRNA expression of genes involved in extracellular matrix assembly were heightened from day 7 to day 21 after coronary occlusion.

We are aware that cardiac ECM is a complex network and alterations in the mRNA expression of other interstitial components (i.e. laminin or fibronectin as well as the cellular component) are also of interest. However, the amount of data derived from this meta-analysis is huge and, for that reason, we decided to focus on alterations in collagen fibers, MMPs and TIMPs.